

# Integration of lncRNA–miRNA–mRNA reveals novel insights into oviposition regulation in honey bees

Xiao Chen[1], Ce Ma[2], Chao Chen[1], Qian Lu[2], Wei Shi[1], Zhiguang Liu[1], Huihua Wang[1] and Haikun Guo[1]

[1] Institute of Apicultural Research, Chinese Academy of Agricultural Sciences, Beijing, China
[2] Novogene Co., LTD, Tianjin, China

## ABSTRACT

**Background.** The honey bee (*Apis mellifera*) is a highly diverse species commonly used for honey production and pollination services. The oviposition of the honey bee queen affects the development and overall performance of the colony. To investigate the ovary activation and oviposition processes on a molecular level, a genome-wide analysis of lncRNAs, miRNAs and mRNA expression in the ovaries of the queens was performed to screen for differentially expressed coding and noncoding RNAs. Further analysis identified relevant candidate genes or RNAs.

**Results.** The analysis of the RNA profiles in different oviposition phase of the queens revealed that 740 lncRNAs, 81 miRNAs and 5,481 mRNAs were differently expressed during the ovary activation; 88 lncRNAs, 13 miRNAs and 338 mRNAs were differently expressed during the oviposition inhibition process; and finally, 100 lncRNAs, four miRNAs and 497 mRNAs were differently expressed during the oviposition recovery process. In addition, functional annotation of differentially expressed RNAs revealed several pathways that are closely related to oviposition, including hippo, MAPK, notch, Wnt, mTOR, TGF-beta and FoxO signaling pathways. Furthermore, in the QTL region for ovary size, 73 differentially expressed genes and 14 differentially expressed lncRNAs were located, which are considered as candidate genes affecting ovary size and oviposition. Moreover, a core set of genes served as bridges among different miRNAs were identified through the integrated analysis of lncRNA-miRNA-mRNA network.

**Conclusion.** The observed dramatic expression changes of coding and noncoding RNAs suggest that they may play a critical role in honey bee queens' oviposition. The identified candidate genes for oviposition activation and regulation could serve as a resource for further studies of genetic markers of oviposition in honey bees.

## INTRODUCTION

The honey bee (*Apis mellifera*) is a highly diverse species commonly used for honey production and pollination services. The oviposition of the honey bee queen is a complex behavior, and it is crucial for the reproductive success and affects the development of the colony (*Woodward, 2010*). However, most reproductive traits are complex in terms of their genetic architecture, present low heritability and are sex-limited (*Manfredini et al.,*

Corresponding author
Wei Shi, xiaochen1984@cau.edu.cn

*2015*; *Mello et al., 2014*). Thus, it is hard to introduce improvements by using traditional selection methods, eg., selective breeding. With the development of molecular technologies, new approaches applied to improve reproductive traits and other complex traits, such as marker-assisted selection (MAS) and genomic selection (*Kramarenko et al., 2014*; *Spötter et al., 2012*). These methods have been widely used in domestic animals for years. However, in honey bees these strategies became popular only in recent years (*Spötter et al., 2012*). A better understanding of the genetic architecture of honey bee will help scientists develop a better strategy for acceleration of the genetic improvement of the reproductive traits.

Honey bees provide an excellent model for oviposition molecular studies. The fact that queens specialize in oviposition, leaving other tasks, for example brood caring, to sterile female workers (*Koeniger, 2008*), potentially reduces the complexity of studying reproductive traits. The process of queens' ovary activation is so fast that queens start to lay eggs around three days after the mating (*Gary, 1992*). In addition, the activity of queens' oviposition is constantly adjusted throughout the year in order to change the colony's strength according to the environmental conditions (*Schneider, 1992*). Such adjustments can be accomplished within a short period (*Koeniger, 2008*), which guarantees colonies' survival and development in the context of dramatic changes of internal and external conditions. Molecular studies have shown that these changes and regulations are associated with profound differences in coding gene expressions (*Lago et al., 2016*; *Pandey & Bloch, 2015*) such as ecdysone receptor (*EcR*), mushroom body large-type Kenyon cell-specific protein-1 (*MBLK-1*), ecdysone-induced protein 74 (*E74*) and ultraspiracle (*Usp*) (*Pandey & Bloch, 2015*).

Recently, the characterization of non-coding RNAs, microRNAs (miRNAs) and long non-coding RNAs (lncRNAs) has become a fruitful area of animals and plants researches. In previous works, several miRNAs, such as bantam, miR-184 and miR-315, have been reported to play important roles in modulating tissue patterns, cell differentiation, ovary development and caste determination in honey bees (*Ashby et al., 2016*; *Macedo et al., 2016*). Additionally, miR-14 and miR-8 have been suggested to be associated with juvenile hormones (*JH*) and ecdysteroids (*Ec*), which play key roles in ovary development and other reproductive behaviors in honey bees (*Boecking, Bienefeld & Drescher, 2000*; *Flatt, Tu & Tatar, 2005*; *Goodman & Cusson, 2012*; *Hartfelder & Emlen, 2005*; *Hoover et al., 2003*; *Riddiford, 1994*; *Wyatt & Davey, 1996*). The other highly expressed non-coding RNAs, lncRNAs, also have a great influence in biological processes, such as cell differentiation, development, immune responses and tumourigenesis (*Okazaki et al., 2002*; *Ota et al., 2004*; *Wilusz, Sunwoo & Spector, 2009*). Moreover, *Necsulea et al. (2014)* found lncRNAs that were preferentially expressed in animals' ovary, and lincRNAs (long intergenic non-coding RNAs) were observed by *Jayakodi et al. (2015)* in *Apis mellifera* to be preferentially expressed in ovary tissue. Furthermore, lncRNAs can be targeted by miRNAs and thus regulate the expression of mRNAs (*Fan et al., 2015*; *Gong et al., 2016*). Therefore, it is valuable to investigate the critical role of lncRNAs, miRNAs and the lncRNA-miRNA-mRNA network in honey bee queens' oviposition.

In order to identify differentially expressed RNAs in ovary activation and oviposition regulation process, we first examined the lncRNA, miRNA and mRNA expression profiles in

the ovaries of virgin queens, egg-laying queens, egg-laying inhibited queens and egg-laying recovered queens using high throughput sequencing method, then compared the RNA expression patterns to help identify candidate genes and/or RNAs that contribute to oviposition activation and regulation. Next, we selected candidate genes or RNAs which may have high effects in regulating ovary size and oviposition by assign the differently expressed RNAs into a QTL for ovary size. Furthermore, the lncRNA-miRNA-mRNA network was constructed to explore the interaction among different RNAs.

## MATERIALS AND METHODS

### Ethics statement

The apiaries for honey bee sample collection were maintained by Institute of Apicultural Research, Chinese Academy of Agricultural Sciences (IAR, CAAS), Beijing, China. No specific permits were required for the described studies.

### Sampling

All samples were obtained from *Apis mellifera ligustica* honeybee colonies. In June 2015, 20 sister queens from a single source colony were reared using standard beekeeping techniques (*Harbo, 1986*). Five days before the emergence, the queens were transferred to an incubator at 36 °C and kept individually in plastic vials. One day old, the queens were marked and each was introduced to her own nucleus colony. The strength of each colony was similar. The entrance of each hive was covered with a queen excluder that confined the queen within the hive but allowed workers to exit and enter.

   Six day old queens were randomly assigned to one of the four groups representing different treatments: (1) virgin queens ($n = 3$); (2) egg-laying queens ($n = 3$) that successfully laid eggs after instrumental insemination; (3) egg-laying inhibited queens ($n = 3$) consisting of egg-laying queens caged in a small cage and kept inside the original hive for seven days; (4) egg-laying recovery queens ($n = 3$), which were first caged in a small cage inside the original hive for seven days to prevent them from egg-laying and then released into their individual colonies for 24 h. All egg-laying recovery queens were able to lay eggs within the 24 h after their release from the small cages. Ovaries of all queens in the four groups were extirpated and stored at −80 °C at the end of the treatment. For instrumental insemination, the source and quantity of the semen was the same for all mated queens. Each sample consisted of the ovary from a single queen. Three samples per treatment group were used for RNAseq (total = 12 samples).

### RNA extraction and library preparation for sequencing

Total RNA was extracted from ovary samples using Trizol reagent (Invitrogen, Carlsbad, CA, USA) according to the manufacturer's instructions. The purity of RNA was checked using the NanoPhotometer spectrophotometer (Implen, Westlake Village, CA, USA), and the concentration was measured using Qubit RNA Assay Kit in Qubit 2.0 Flurometer (Life Technologies, Carlsbad, CA, USA). The integrity of RNA was assessed using the RNA Nano 600Assay Kit of the Agilent Bioanalyzer 2100 system (Agilent Technologies, Santa Clara, CA, USA).

LncRNA and mRNA library preparation was carried out using NEBNext® Ultra™ Directional RNA Library Prep Kit for Illumina® (NEB, Ipswich, MA, USA) following manufacturer's recommendations. Paired-end reads of 150 bp were generated using the Illumina Hiseq 4000 platform. After quality control, paired-end clean reads were aligned to the reference genome (*Amel_4.5*) using TopHat v2.0.9. Transcripts were assembled and annotated using Cufflinks (http://cufflinks.cbcb.umd.edu/). The known mRNAs and lncRNAs were identified according to the annotation of *Apis mellifera* genome sequence (*Amel_4.5*). The remaining transcripts were used to screen for putative lncRNAs using the following criteria: (1) length $\geq$ 200 bp; (2) exon number $\geq$ 2; (3) sequencing coverage $\geq$ 3; (4) identified in at least two samples. The transcripts meeting the above criteria were further filtered by removing known non-lncRNA transcripts. Then, the transcripts that passed the filters were evaluated for coding potential using CPC (0.9-r2) (*Kong et al., 2007*) and Pfam-scan (v1.3) (*Punta et al., 2012*). Only those without coding potential were categorized as novel lncRNAs.

Small RNA library preparation was carried out using NEBNext® Multiplex Small RNA Library Prep Set for Illumina® (NEB, Ipswich, MA, USA) following manufacturer's recommendations. Single-end reads of 50 bp were generated using the Illumina Hiseq 2500 platform. After quality control, the clean reads were mapped to reference sequence (*Amel_4.5*) applying Bowtie (*Langmead et al., 2009*). Mapped reads were used to identify known miRNAs using miRBase 20.0 (*Griffiths-Jones, 2010*). Novel miRNAs were predicted with miREvo (*Ming et al., 2012*) and mirdeep2 (*Friedländer et al., 2012*) through exploring the characteristic hairpin structure, Dicer cleavage sites and minimum free energy.

All the sequencing data are available through the GEO database with accession number GSE93028.

## Differentially expressed lncRNAs, miRNAs and mRNAs identification and clustering analysis

Differentially expressed (DE) lncRNAs, miRNAs and mRNAs (Benjamini & Hochber method corrected *p*-value <0.05) were identified using DESeq R package (1.8.3) for each of the following comparisons: (1) egg-laying queens *vs.* virgin queens (ovary activation process); (2) egg-laying inhibited queens *vs.* egg-laying queens (oviposition inhibition process); (3) egg-laying recovery queens *vs.* egg-laying inhibited queens (oviposition recovery process). Furthermore, the expression of each RNA type was analyzed with unsupervised hierarchical clustering with the R package of "pheatmap". To do unsupervised hierarchical clustering, firstly, the expression of RNA was normalized. For normalization of lncRNA and mRNA, the following formula was used: $FPKm = \log_{10} FPKM + 1$. For normalization of miRNA, the following formula was applied: $TPm = \log_{10} TPM + 1$ (TPM, transcripts per kilobase million). Then the Euclidean distance was used to measure the degree of similarity between the expression profiles of samples. The method in the package to cluster distance is "complete".

## Prediction of lncRNA and miRNA target genes

The potential trans role of lncRNAs (acting on non-neighboring genes) can be assessed by correlating expression levels between lncRNAs and mRNAs. The trans role of lncRNAs

in coding genes was examined based on the expression correlation coefficient (Pearson correlation $\geq 0.95$ or $\leq -0.95$). To predict miRNAs targets, we searched for the targets in the 3′ UTR of genes models. For genes lacking a predicted 3′ UTR, the region 1,000 bp downstream of the stop codon were included. The prediction was performed by Miranda with the following parameter: free energy $< -10$ kcal/mol and score $>140$ (*Enright et al., 2003*).

## Functional enrichment analysis

*Apis mellifera* gene set was annotated based on the corresponding *Drosophila melanogaster* orthologues and catergorized by their biological functions. Gene annotation was done by a homology-based method. *Apis mellifera* CDS sequences were blasted against the *Drosophila melanogaster* peptide sequences (Ensembl database *Release 74*) using the comment "-p blastx -m8 -e 1e-5-F F". The minimum peptide alignment must be more than 50 aa. The correspondence relationship of *Apis mellifera* genes and ontology categories was decided by the hit with the best alignment score. Gene ontology (GO) enrichment analysis with the *Drosophila melanogaster* reference gene set was implemented by the GOseq R package (*Young et al., 2012*). KEGG pathways analysis was performed using KOBAS to determine the involvement of genes in different biological pathways (*Mao et al., 2005*).

## Chromosomal localization of DE lncRNAs and mRNAs in quantitative trait locus (QTL) for ovary Size

The localization of the DE lncRNAs and DE mRNAs on *Apis mellifera* chromosomes was accessed from NCBI database (*Amel_4.5*). Each RNA location was estimated in centimorgans and was compared with the location of a significant QTL previously identified for ovary size. This QTL locates on chromosome 11 between the position 8.9 Mb and 12.2 Mb (*Graham et al., 2011*; *Linksvayer et al., 2009*). Genes or RNAs which locate within the QTL confidence intervals were accepted as candidate genes for ovary size and potential candidate genes for oviposition.

## Construction of lncRNA-miRNA-mRNA network

To construct lncRNA-miRNA-mRNA network, we first selected lncRNAs which were predicted to act as miRNA targets or decoys by Fan's methods (*Fan et al., 2015*). Next, to define the miRNA-mRNA relationships, the Pearson correlation coefficient value between a miRNA and its target mRNA was calculated, and strongly correlated miRNA-mRNA pairs (the absolute value of greater 0.8) were selected (either positive or negative). To construct the network, each DE RNA node must be either in a lncRNA-miRNA pair or in a miRNA-mRNA pair. The nodes in the network consisted of miRNAs, lncRNAs acting as miRNA targets, lncRNAs acting as miRNA decoys, mRNAs acting as miRNA targets. The network was visualized using Cytoscape (version 3.4.0) (*Smoot et al., 2011*).

## Real time PCR

In order to confirm sequencing results, the expression of five lncRNAs, five mRNAs and five miRNAs were validated by real time PCR using the same 12 ovary samples used for sequencing. Following total RNA extraction, ovarian samples were reversely transcribed to
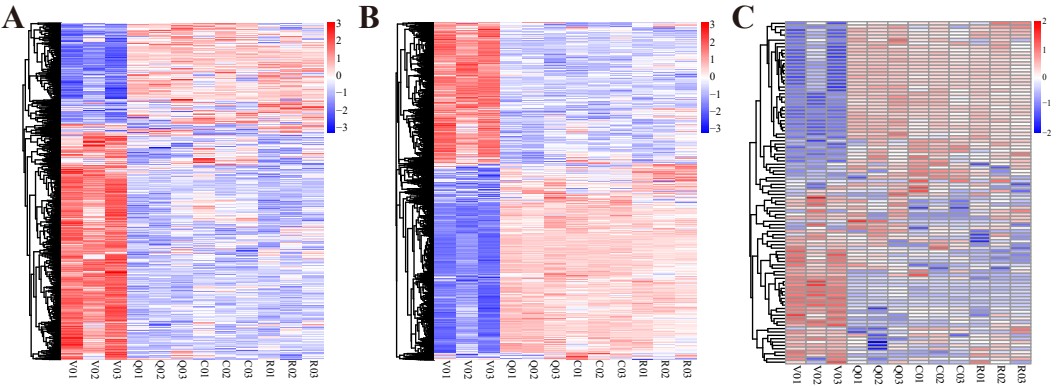

**Figure 1** **The cluster heat map of expression profiles of lncRNAs, mRNAs and miRNAs at different statuses during ovary activation and oviposition regulation.** (A) The cluster heat map of expression profiles of lncRNAs; (B) the cluster heat map of expression profiles of mRNAs; (C) the cluster heat map of expression profiles of miRNAs.V, group of virgin queens ($n = 3$); Q, group of egg-laying queens ($n = 3$); C, group of egg-laying inhibited queens ($n = 3$); R, group of egg-laying recovery queens ($n = 3$).

generate cDNA. For cDNA synthesis of lncRNA and mRNA, an M-MLV FIRST STRAND KIT (Invitrogen, Shanghai, China) and an oligo (dT)18 primer were used in a reverse transcription reaction of 20 µl, following the supplier's instructions. For miRNA cDNA synthesis, a miRcute miRNA cDNA synthesis kit (Tiangen Biotech, Beijing, China) was used. In brief, *E.coli* Poly(A) Polymerase was used to add a poly(A) tail at 3′ end and then Oligo(dT)-Universal tag was used in a reverse transcription reaction following the supplier's instructions. Two microliters of each cDNA was subjected to PCR amplification using specific primers (Table S1). The PCR efficiency of each gene was estimated by standard curve calculation using four points of cDNA serial dilutions. Cycle threshold ($Ct$) values were transformed to quantities using the comparative $Ct$ method, setting the relative quantities of the virgin queens group for each gene to 1 (quantity $= 10 - \Delta Ct/\text{slope}$). Data normalization of lncRNA and mRNA were carried out using the Actin reference gene. Data normalization of miRNA was carried out using the U6 reference gene. The correlation between the results of sequencing and PCR was calculated using correlation test.

## RESULTS

### Genome-wide identification of DE lncRNAs, mRNAs and miRNAs from honey bee queens

Sequencing of all lncRNA and mRNA libraries generated 1,243,644,174 raw paired-end reads with a length of 150 bases, resulting in a total of 16.7 gigabases. Sequencing of all miRNA libraries generated 152,659,565 raw single-end reads with a length of 50 bases, resulting in a total of 7.631 gigabases. The whole expression profiles of lncRNAs, miRNAs and mRNAs of ovaries at four different conditions are presented in Fig. 1. From the expression profiles, DE lncRNAs, mRNAs and miRNAs were discriminated between different groups (Table 1 and Table S2). A total of 740 lncRNAs, 5481 mRNAs and 81 miRNAs were differentially expressed in ovary activation process (egg-laying queens *vs.* virgin queens). Eighty-eight lncRNAs, 338 mRNAs and 13 miRNAs were

**Table 1   Number of differentially expressed coding and non-coding RNAs identified from each comparison.**

| Number of differentially expressed RNAs | Ovary activation (Egg-laying queens compared with virgin queens) | | Oviposition inhibition (Egg-laying inhibited queens compared with egg-laying queens) | | Oviposition recovery (Egg-laying recovery queens compared with egg-laying inhibited queens) | |
|---|---|---|---|---|---|---|
| | Up-regulated | Down-regulated | Up-regulated | Down-regulated | Up-regulated | Down-regulated |
| mRNAs | 3218 | 2263 | 266 | 72 | 256 | 241 |
| lncRNAs | 224 | 516 | 57 | 31 | 40 | 60 |
| miRNAs | 39 | 42 | 9 | 4 | 2 | 2 |

differentially expressed in oviposition inhibition process (egg-laying inhibited queens *vs.* egg-laying queens). One hundred lncRNAs, 497 mRNAs and four miRNAs were differentially expressed in oviposition recovery process (egg-laying recovery queens *vs.* egg-laying inhibited queens). A summary of the up-/down-regulated information is shown in Table 1.

## GO and pathway enrichment analysis

Functional annotation analysis of target genes of the DE lncRNAs, miRNA and mRNA was performed to identify GO terms and KEGG pathways with higher confidence (Table S3–S5). Because GO terms and pathways enriched with the DE lncRNAs, miRNA and mRNAs were similar to each other, here we only describe the enrichment results of DE mRNAs. In the ovary activation process, most of the enriched GO_BP terms of DE mRNAs were involved in tissue development, energy producing and hormone biosynthesis and metabolism, such as oocyte microtubule cytoskeleton polarization, fatty acid oxidation, neurotrophin signaling pathway, ecdysteroid catabolic process (Table S3). In the oviposition inhibition process, contrary to the ovary activation process, several GO terms were not enriched, but enrichment occurred again when oviposition recovered, such as cellular response to transforming growth factor beta stimulus, positive regulation of cyclase activity, post-embryonic hemopoiesis, larval lymph gland hemopoiesis, eye pigment biosynthetic process, and compound eye cone cell fate commitment (Table S3).

DE mRNAs enrichment ($p < 0.05$) was seen in KEGG pathways (Table S3). Several pathways were both enriched in ovary activation and oviposition regulation process, namely glycerolipid metabolism, glycerophospholipid metabolism, hippo signaling pathway—fly, inositol phosphate metabolism, MAPK signaling pathway—fly, neuroactive ligand–receptor interaction, notch signaling pathway, phosphatidylinositol signaling system and Wnt signaling pathway (Table 2).

## Chromosomal localization of DE lncRNAs and mRNAs in QTL region for ovary size

If the differentially expressed lncRNAs and mRNAs were found located within the confidence interval of the QTL for ovary size, they could be regarded as candidate genes for ovary size and potential candidate genes for oviposition. In this way, 73 candidate genes and 14 lncRNAs (Table S6) were identified.
**Table 2 Intersection set of significantly enriched pathways with DE lncRNAs, DE mRNAs and DE miRNAs.**

| Significantly enriched pathways | Q_V | C_Q | R_C | Significantly enriched pathways | Q_V | C_Q | R_C |
|---|---|---|---|---|---|---|---|
| Arginine and proline metabolism |  | ✓ | ✓ | mTOR signaling pathway | ✓ | ✓ | ✓ |
| Base excision repair | ✓ |  | ✓ | Mucin type O-Glycan biosynthesis | ✓ |  |  |
| Biosynthesis of amino acids | ✓ | ✓ |  | Neuroactive ligand–receptor interaction | ✓ | ✓ | ✓ |
| Circadian rhythm - fly | ✓ | ✓ | ✓ | N-Glycan biosynthesis | ✓ | ✓ |  |
| Cysteine and methionine metabolism | ✓ | ✓ |  | Nitrogen metabolism | ✓ | ✓ |  |
| Dorso-ventral axis formation | ✓ | ✓ | ✓ | Notch signaling pathway | ✓ | ✓ | ✓ |
| Drug metabolism - other enzymes | ✓ | ✓ |  | Other glycan degradation |  |  |  |
| ECM-receptor interaction | ✓ | ✓ |  | Peroxisome | ✓ |  |  |
| Endocytosis | ✓ | ✓ |  | Phenylalanine metabolism | ✓ | ✓ |  |
| Folate biosynthesis |  | ✓ |  | Phosphatidylinositol signaling system | ✓ | ✓ | ✓ |
| FoxO signaling pathway | ✓ | ✓ | ✓ | Phototransduction - fly | ✓ | ✓ | ✓ |
| Galactose metabolism | ✓ |  |  | Proteasome | ✓ | ✓ |  |
| Glycerolipid metabolism | ✓ | ✓ | ✓ | Protein processing in endoplasmic reticulum |  | ✓ |  |
| Glycerophospholipid metabolism | ✓ | ✓ | ✓ | Purine metabolism | ✓ | ✓ | ✓ |
| Glycosaminoglycan biosynthesis - chondroitin sulfate / dermatan sulfate |  | ✓ |  | Retinol metabolism |  | ✓ |  |
| Glycosaminoglycan biosynthesis - heparan sulfate / heparin |  | ✓ | ✓ | RNA degradation | ✓ | ✓ | ✓ |
| Hedgehog signaling pathway | ✓ | ✓ |  | RNA transport | ✓ | ✓ | ✓ |
| Hippo signaling pathway - fly | ✓ | ✓ | ✓ | Spliceosome | ✓ | ✓ | ✓ |
| Inositol phosphate metabolism | ✓ | ✓ | ✓ | Starch and sucrose metabolism | ✓ |  |  |
| Jak-STAT signaling pathway |  |  | ✓ | Sulfur metabolism | ✓ |  |  |
| Lysine degradation | ✓ |  |  | TGF-beta signaling pathway | ✓ | ✓ | ✓ |
| MAPK signaling pathway - fly | ✓ | ✓ | ✓ | Tyrosine metabolism | ✓ | ✓ |  |
| Metabolic pathways | ✓ | ✓ | ✓ | Ubiquinone and other terpenoid-quinone biosynthesis | ✓ | ✓ |  |
| mRNA surveillance pathway | ✓ | ✓ | ✓ | Wnt signaling pathway | ✓ | ✓ | ✓ |

**Notes.**

V, group of virgin queens; Q, group of egg-laying queens; C, group of egg-laying inhibited queens; R, group of egg-laying recovery queens.

## Construction of the lncRNA-miRNA-mRNA network

The bioinformatic analysis predicted that 469 lncRNAs were targeted by 69 miRNAs and 117 lncRNAs acted as decoys to 31 miRNAs. The transcriptome network was constructed based on the lncRNA-miRNA and the miRNA-mRNA relationship pairs. The resulting network consists of 229 lncRNA-miRNA pairs and 225 miRNA-mRNA pairs (Fig. S1 and Table S7).

To further investigate the potential candidate genes and RNAs for ovary activation and oviposition, a reproductive associated network was constructed containing the DE miRNAs and mRNAs which played specific or suspected roles in reproduction, and the DE lncRNAs and mRNAs located in the QTL region for ovary size. The network was constructed with 105 lncRNA-miRNA pairs and 83 miRNA-mRNA pairs, consisted of 105 lncRNAs, 25 miRNAs and 74 mRNAs (Fig. 2 and Table S8).

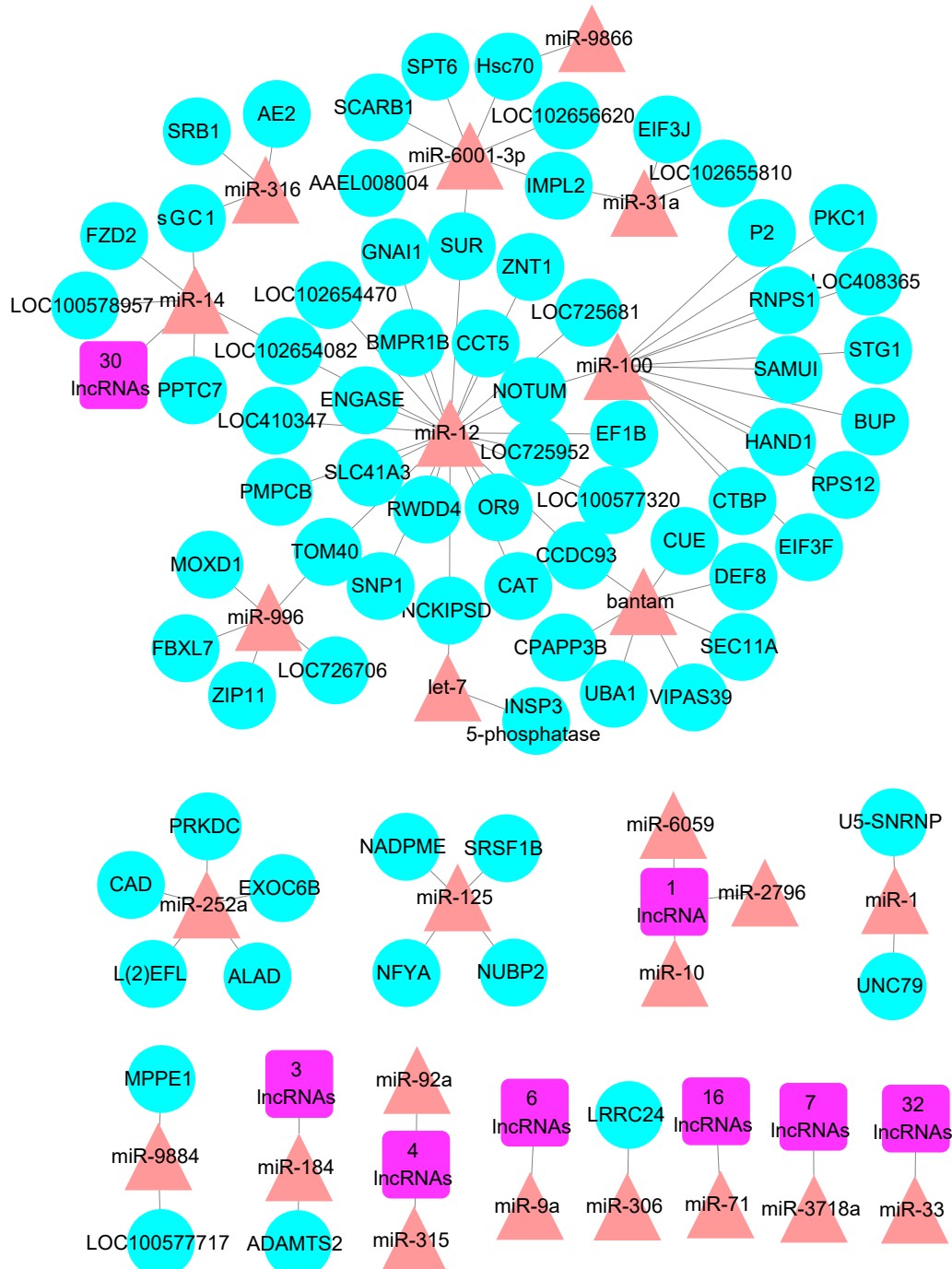

**Figure 2** **The reproductive associated lncRNA-miRNA-mRNA network.** The network was constructed with DE lncRNAs, DE miRNAs and DE mRNAs which have known or suspected roles in reproduction and/or located in the QTL region for ovary size. Purple square nodes represent lncRNAs. Red triangle nodes represent miRNAs. Blue circle nodes represent mRNAs.

## Validation of RNA-Seq data by real time PCR

In order to validate the sequencing results, the expression of five lncRNAs, five mRNAs and five miRNAs were tested by using real time PCR with the same RNA samples used for sequencing (Table S1). The expression profiles of these genes/RNAs detected by real time PCR were consistent with those obtained by sequencing, which confirmed the reliability of our sequencing results.

## DISCUSSION

In the present study, dynamical lncRNAs, mRNAs and miRNAs expression profiles in ovary activation and oviposition processes in honey bees were identified. However, the complex molecular mechanism behind the oviposition activation and regulation still needs to be illustrated.

### Representative enriched pathways

The gene function analysis showed that DE RNAs enrichment was seen in a number of pathways in ovary activation and/or oviposition regulation process. Some of the pathways are particularly interesting, such as Wnt, hippo, TGF-beta, notch, MAPK, FoxO and mTOR signaling pathways (Fig. 3). More than 50% of the genes in those pathways were differently expressed according to our results. Some of the pathways have known or suspected roles in honey bees. For example, Wnt, hippo, notch, MAPK and TOR pathways were reported to be involved in caste determination in honey bees (*Ashby et al., 2016*; *Wheeler, Buck & Evans, 2014*). Caste determination is inseparably linked with the ovary development status. Although, so far, studies on the effect of these pathways on oviposition are not available, some insights can be drawn from other species.

The Wnt signaling pathway was found to be involved in the development of reproductive system such as the development of ovarian follicles, ovulation and luteinization (*Sun & Wang, 2003*). The hippo signaling pathway was also reported to be related to the regulation of mouse ovarian functional remodeling (*Ye et al., 2017*). Moreover, the hippo signaling pathway can coordinate with Wnt, TGF-beta and notch signaling pathways affecting organ size in *Drosophila* (*Barry & Camargo, 2013*). Because after queen mating, the size of ovary will become bigger than the virgin's (*Koeniger, 2008*), we also observed many genes in Wnt, TGF-beta, hippo and notch signaling pathways that were differentially expressed in mated queens compared with virgin queens. It indicated that those pathways may participate in ovarian function remodeling after mating to prepare for oviposition in honey bees. The oocyte growth and development is crucial to successful oviposition, particularly during the height of the brood-rearing season when a good queen can lay up to 1,500 eggs per day (*Koeniger, 2008*). Studies in mammal found that TGF-beta, MAPK and FoxO signaling pathways regulate oocytes growth and development (*Edmonds et al., 2010*; *Kretzschmar, Doody & Massagué, 1997*; *Zhang et al., 2011*). Also, there were studies showing that the TGF-beta signaling pathway was essential for oogenesis in *Drosophila* (Twombly et al., 1996). TGF-beta, MAPK and FoxO signaling pathways demonstrated enrichment in DE RNAs in our results, which indicated that these pathways may involve in oocyte growth and development in honey bees.

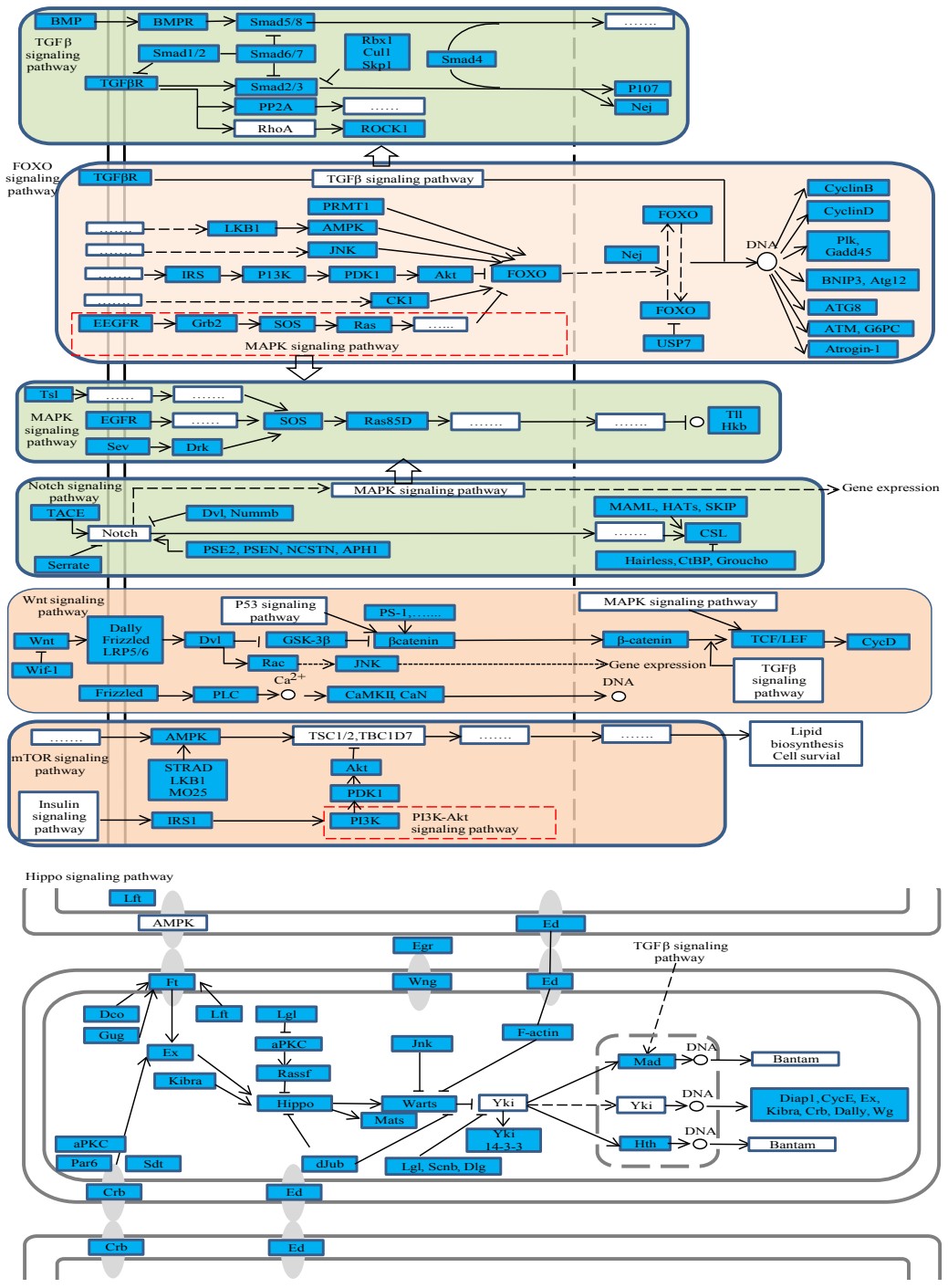

**Figure 3 The representative enriched pathways map.** DE genes were marked in blue. Genes without color and "......" stand for genes involved in the pathway but not differentially expressed in our results.

The queen is the only fertile female in a honey bee colony, and it constrains the reproduction of worker bees. A recent study reported that notch signaling facilitated the queen to repress ovary activity and maintain reproductive sterility in the worker bees (*Duncan, Hyink & Dearden, 2016*). Also, TOR pathway was found to be associated with the reproductive status in workers (*Patel et al., 2007*). DE RNAs enrichment was observed in the present study in both notch and TOR signaling pathways in mated queens, which demonstrated that notch and TOR pathways possessed signaling functions in strengthening the reproductive constraint after queen mating.

Further, the studied pathway maps were looked up in KEGG database to assess whether there is a relationship among them. The results showed that they were closely interacting with each other as shown in Fig. 3, whereby, for example, TGF-beta signaling pathway was part of hippo and Wnt signaling pathways. These pathways were enriched both in oviposition activation and oviposition process. Considering the roles of these pathways in ovarian function remodeling, oocyte growth and development and other related processes, they are critical for a successful oviposition by complex fine-tuning relationships.

Among the DE genes in those pathways, several genes were found to participate in more than one pathway. The gene nejire (*Nej*, also known as CREB-binding protein (*CBP*)) participated in three pathways, namely notch, FoxO and TGF-beta signaling pathways. Additionally, *Nej* was significantly up-regulated in the egg-laying queens compared to virgin queens. Also, studies in *Drosophlia melanogaster* found that *Nej* was involved in regulation of many pathways during embryo development, through hedgehog, wingless and TGF-beta signaling pathways (*Fernandez-Nicolas & Belles, 2016*). Taken together, we could conclude that *Nej* may participate in embryonic development in honey bees through notch, FoxO and TGF-beta signaling pathways, and can be considered as the potential candidate genes for oviposition.

## Genes and lncRNAs co-localized in QTL region for ovary size

We compared the location of the DE genes and DE lncRNAs on the honey bee genome available at the NCBI database (*Amel_4.5*) with one QTL for ovary size. Seventy-three genes and 14 lncRNAs were identified, and some of them together with their key function will be explained further.

Among the 73 genes, G2/mitotic-specific cyclin-B3 (*CycB3*) is the one we paid special attention. It was shown for example that *CycB3* controlled oocyte maturation and early embryo development in mouse (*Polański, Homer & Kubiak, 2012*), but studies of *CycB3* in reproduction in honey bees are scarce. Figure 3 showed that *CycB3* was significantly up-regulated in ovary activation process and participated in the FoxO signaling pathway, which implies that *CycB3* may play important roles in oviposition and affect oocyte maturation in honey bees through FoxO signaling pathway.

Two lncRNAs, XLOC_073978 and XLOC_081294 (sequence information noted in Table S2) are of particular interest. The predicted targets of XLOC_073978 included myophilin-like, yellow-f and cytochrome P450 9Q1 (*CYP9Q1*). The predicted targets of XLOC_081294 included yellow-b, odorant binding protein 10 (*Obp10*), myosin regulatory light chain 2 (*Mlc2*), *CYP9Q1, CYP9Q2* and *CYP9Q3*. Myophilin (also known as *CHD64*) was previously

identified as *JH* response gens (*Rewitz et al., 2006*), which regulated many aspects of physiology and development of insects (*Flatt, Tu & Tatar, 2005*), including reproduction (*Flatt, Tu & Tatar, 2005*; *Goodman & Cusson, 2012*; *Hartfelder & Emlen, 2005*; *Riddiford, 1994*; *Wyatt & Davey, 1996*). *Mlc2* was previously detected changing expression during the ovary activation process (*Manfredini et al., 2015*). Concerning yellow-b, yellow-f and *Obp10*, they had been reported to relate to ovary activation and response with *Ec*, one of the most critical hormones affecting reproduction in honey bees and other insects (*Amdam et al., 2010*; *Bloch, Hefetz & Hartfelder, 2000*; *Hagedorn, 1985*; *Pandey & Bloch, 2015*). Furthermore, it is notable that three target genes from CYP450 family (*CYP9Q1, CYP9Q2* and *CYP9Q3*), some of which were previously detected to interact with *Ec* (*Mello et al., 2014*; *Rewitz et al., 2006*), showed changes of expressions in our results.

Therefore, all the predicted targets of XLOC_073978 and XLOC_081294 were associated with reproduction in honey bees. This highlights their roles in oviposition. Because the genes elsewhere in the genome might share pathways with genes in the QTL region and reflect downstream effects of the QTL (*Fernandez-Rodriguez et al., 2011*), they will be useful for identifying candidate genes and/or RNAs for ovary activation and oviposition by combining the information obtained from expression analysis with the QTL location analyses. Further studies will involve in studying the genes that interacted with the QTL genes.

### Analysis of DE RNAs with known or suspected roles in reproduction

Tables 3 and 4 show that 31 mRNAs and 36 miRNAs were significantly regulated in caste determination or other reproductive related process, which indicated that they have known or suspected roles of in ovary activation. The oviposition status is positively correlated with two genes (heat shock protein 90 (*Hsp90*) and *Usp*) and negatively correlated with ten genes. *Hsp90* has been reported as a candidate marker gene for caste-specific ovary development (*Lago et al., 2016*). According to our results, *Hsp90* can also be a candidate marker for the oviposition status of the honey bee queen. Among the negatively correlated genes, four are CYP450 family genes. Several genes of CYP450 family were reported to act as response genes of *Ec* and 20-hydroxyecdysone (*20E*) which is the active *Ec* in most insects (*Buszczak & Segraves, 1998*) including honey bees (*Yamazaki et al., 2011*). Importantly, some CYP450 genes were identified as targets of lncRNAs, which are located in the QTL region of ovary size in our results. This highlights their roles in oviposition.

The other six miRNAs that are negatively correlated with the oviposition status are bantam, miR-12, miR-279a-3p, miR-31a, miR-993 and miR-996. Bantam plays an important role in embryonic development and was identified as a crucial target of the signaling pathways of hippo and EGFR/MAPK in *Drosophila* (*Herranz, Hong & Cohen, 2012*; *Nolo et al., 2006*; *Thompson & Cohen, 2006*). The DE mRNAs enrichment was seen in hippo and EGFR/MAPK signaling pathways in our study, suggesting that bantam may affect ovary activation or oviposition by the hippo and/or EGFR/MAPK signaling pathway. Also, four miRNAs (miR-1, miR-133, miR-184 and miR-190) were down-regulated during oviposition activation and recovery, but the suspension of oviposition did not affect their expression. MiR-184, which is highly conserved and widely studied in insects, was reported to affect caste determination of honey bees (*Guo et al., 2013*; *Macedo et al., 2016*; *Mello et*

Chen et al. (2017), *PeerJ*, DOI 10.7717/peerj.3881

**Table 3  Analysis of DE genes with known or suspected roles in honey bee reproduction or related process.**

| Gene id | Gene name | Correlate | *Ref.* | Expression level in this study | | |
|---|---|---|---|---|---|---|
| | | | | Q *vs.* V | C *vs.* Q | R *vs.* C |
| 408961 | Apolipophorins (known as *RFABP*) | *JH* response genes | *Pandey & Bloch (2015)* | Down-regulated | Up-regulated | Down-regulated |
| 552515 | ATP-dependent RNA helicase WM6 (known as Helicase at 25E) | Higher expressed in ovaries of queen larvae compared with worker larvae in fourth and early fifth larvae | *Lago et al. (2016)* | Up-regulated | Down-regulated but not significantly | Up-regulated but not significantly |
| 408827 | Carbonic anhydrase 1 (*CAH1*) | *JH* response genes | *Pandey & Bloch (2015)* | Down-regulated | Up-regulated but not significantly | Down-regulated but not significantly |
| 413762 | Complement component 1 Q subcomponent-binding protein, mitochondrial | Higher expressed in ovaries of queen larvae compared with worker larvae in fourth and early fifth larvae | *Lago et al. (2016)* | Down-regulated | Down-regulated but not significantly | Up-regulated |
| 726690 | Cytochrome P450 6AS3 (*CYP6AS3*) | Up-regulated in the *EcR* knock down bees | *Mello et al. (2014)* | Down-regulated | Up-regulated | Down-regulated but not significantly |
| 412209 | Cytochrome P450 6AS4 (*CYP6AS4*) | Up-regulated in the *EcR* knock down bees | *Mello et al. (2014)* | Down-regulated | Up-regulated | Down-regulated |
| 409677 | Cytochrome P450 6AS5 (*CYP6AS5*) | Up-regulated in the *EcR* knock down bees | *Mello et al. (2014)* | Down-regulated | Up-regulated | Down-regulated |
| 551560 | Cytochrome P450 6BD1 (*CYP6BD1*) | Up-regulated in the *EcR* knock down bees | *Mello et al. (2014)* | Down-regulated | Up-regulated | Down-regulated |
| 411057 | Cytochrome P450 314 A1 (*CYP314A1*) | coded for *Ec* dysone 20-hydroxylase | *Rewitz et al. (2006)* | Up-regulated | No change | Down-regulated but not significantly |
| 406143 | Defensin 1 (*Def1*) | Up-regulated in mated queens compared with virgin queens | *Manfredini et al. (2015)* | Down-regulated | Up-regulated | Down-regulated |
| 406070 | Dopamine receptor 2 (*Dopr2*) | *Ec* response genes | *Rewitz et al. (2006)* | Down-regulated | No change | Up-regulated |
| 410309 | *Ec* dysone-induced protein 75 (*E75*) | *Ec* response genes | *Rewitz et al. (2006)* | Up-regulated | Up-regulated but not significantly | No change |
| 406084 | *Ec* dysone receptor (*EcR*) | *Ec* response genes | *Rewitz et al. (2006)* | Up-regulated | Not detected | Not detected |
| 408758 | *Ec* dysteroid-regulated gene E74 (*E74*) | *Ec* response genes | *Rewitz et al. (2006)* | Up-regulated | No change | No change |
| 409384 | Heat shock protein 60 (*Hsp60*) | Higher expressed in ovaries of queen larvae compared with worker larvae in fourth and early fifth larvae | *Lago et al. (2016)* | Up-regulated but not significantly | Down-regulated | Up-regulated |

Chen et al. (2017), *PeerJ*, DOI 10.7717/peerj.3881

**Table 3** (*continued*)

| Gene id | Gene name | Correlate | *Ref.* | Expression level in this study | | |
|---|---|---|---|---|---|---|
| | | | | Q *vs.* V | C *vs.* Q | R *vs.* C |
| 408928 | Heat shock protein 90 (*Hsp90*) | Higher expressed in ovaries of queen larvae compared with worker larvae in fourth and early fifth larvae; candidate marker genes for caste-specific ovary development; | *Lago et al. (2016)* | Up-regulated | Down-regulated | Up-regulated |
| 408818 | Hexokinase (*HK*) | QMP response genes | *Hoover et al. (2003)* | Up-regulated | No change | Down-regulated but not significantly |
| 406117 | Hexamerin 70b (*Hex70b*) | *JH* response gene, highly expressed in fourth and early fifth-instar queen ovaries | *Lago et al. (2016)* | Up-regulated | Up-regulated but not significantly | Down-regulated but not significantly |
| 726542 | Histone H3 | QMP response genes; overrepresented in ovaries of queens in the fifth larval instar | *Humann & Hartfelder (2011)* | Down-regulated | No change | Down-regulated but not significantly |
| 102655073 | Histone H4 | QMP response genes | *Hoover et al. (2003)* | Up-regulated | Up-regulated but not significantly | Down-regulated but not significantly |
| 726965 | *JH*-inducible protein | *JH* and *Ec* response gene, up-regulated in *Ec* knock down bess | *Mello et al. (2014)* | Up-regulated | Down-regulated but not significantly | No change |
| 100576395 | Kruppel homolog 1 (*Kr-h1*) | An immediate response gene in the *JH* response cascade | *Lago et al. (2016)* | Up-regulated | Down-regulated but not significantly | Up-regulated but not significantly |
| 406121 | Major royal jelly protein 3 (Mrjp3) | *Ec* response gene; down-regulated in *Ec* knock down bees | *Mello et al. (2014)* | Down-regulated | Up-regulated but not significantly | Down-regulated but not significantly |
| 409870 | Minor histocompatibility antigen H13 | Higher expressed in ovaries of queen larvae compared with worker larvae in fourth and early fifth larvae | *Lago et al. (2016)* | Up-regulated | Down-regulated but not significantly | No change |
| 411820 | Mitogen-activated protein kinase phosphatase-3 (*Mapk-3*) | Higher expressed in ovaries of queen larvae compared with worker larvae in fourth and early fifth larvae | *Lago et al. (2016)* | Down-regulated | Up-regulated but not significantly | Up-regulated but not significantly |
| 408572 | Myophilin (*CHD64*) | *JH* response genes | *Rewitz et al. (2006)* | Down-regulated | Down-regulated but not significantly | Up-regulated but not significantly |
| 409881 | Myosin regulatory light chain 2 (*Mlc2*) | Up-regulated in mated queens compared with virgin queens | *Manfredini et al. (2015)* | Down-regulated | Down-regulated | Up-regulated but not significantly |
| 552193 | Proton-coupled amino acid transporter | QMP response genes | *Hoover et al. (2003)* | Up-regulated | No change | Down-regulated but not significantly |

(*continued on next page*)

Peer J

**Table 3** (*continued*)

| Gene id | Gene name | Correlate | *Ref.* | Expression level in this study | | |
|---------|-----------|-----------|--------|---------------------------|---|---|
| | | | | Q *vs.* V | C *vs.* Q | R *vs.* C |
| 409681 | RWD domain-containing protein 1 (*RWDD1*) | *QMP* response genes | *Hoover et al. (2003)* | Up-regulated | Down-regulated but not significantly | Down-regulated but not significantly |
| 409227 | Ultraspiracle (*USP*) | *Ec* and *JH* response genes | *Rewitz et al. (2006)* | Up-regulated but not significantly | Down-regulated but not significantly | Up-regulated |
| 406088 | Vitellogenin (*Vg*) | The protein product serves as a yolk precursor in egg development | *Nunes et al. (2013)* | Down-regulated | Up-regulated | Down-regulated |

**Notes.**

V, group of virgin queens ($n = 3$); Q, group of egg-laying queens ($n = 3$); C, group of egg-laying inhibited queens ($n = 3$); R, group of egg-laying recovery queens ($n = 3$).

Chen et al. (2017), *PeerJ*, DOI 10.7717/peerj.3881

**Table 4** Analysis of DE miRNAs with known or suspected roles in honey bee reproduction or related process.

| miRNA id | Correlate | Ref. | Expression level in this study | | |
|---|---|---|---|---|---|
| | | | Q *vs.* V | C *vs.* Q | R *vs.* C |
| Bantam | Caste determination; target of hippo and EGFR/MAPK signaling pathways; critical in embryonic development and the control of cell proliferation and survival; up-regulated in 4-day-old queen larvae compared with 4-day-old worker larvae; related to insulin and Wnt pathway. | *Ashby et al. (2016), Shi et al. (2015)* | Down-regulated | Up-regulated but not significantly | Down-regulated but not significantly |
| Let-7 | Caste determination; major target of steroid pathways; miRNA markers associated with the behavioural shift of worker bees from nurses to forages; immune-related; *Vg* positive correlation; participated in regulation of behavioral maturation in honey bees; associated with reproductive statuses; up-regulated in the inactive ovaries; up-regulated in 4-day-old queen larvae; related to Wnt pathway; down-regulated in *Ec* knock down bees | *Ashby et al. (2016), Macedo et al. (2016), Mello et al. (2014), Nunes et al. (2013), Shi et al. (2015)* | Down-regulated | No change | Up-regulated but not significantly |
| Ame-mir-1 | *Vg* positive correlation; associated with reproductive statuses; up-regulated in the inactive ovaries; down-regulated in *Ec* knock down bees | *Macedo et al. (2016), Mello et al. (2014), Nunes et al. (2013)* | Down-regulated | No change | Down-regulated but not significantly |
| Ame-mir-10 | Up-regulated in 4-day-old queen larvae | *Shi et al. (2015)* | Up-regulated | Down-regulated | Up-regulated but not significantly |
| Ame-mir-100 | 20E and JH response miRNA; caste determination; associated with reproductive statuses; up-regulated in the inactive ovaries | *Ashby et al. (2016), Macedo et al. (2016)* | Down-regulated | Down-regulated but not significantly | Up-regulated but not significantly |
| Ame-mir-12 | Associated with reproductive statuses; up-regulated in 4-day-old queen larvae compared with 4-day-old worker larvae; related to insulin and MAPK pathway; down-regulated in *Ec* knock down bees | *Macedo et al. (2016), Mello et al. (2014), Shi et al. (2015)* | Down-regulated but not significantly | Up-regulated | Down-regulated but not significantly |
| Ame-mir-125 | 20E and JH response miRNA; caste determination; up-regulated in the inactive ovaries; up-regulated in 4-day-old queen larvae compared with 4-day-old worker larvae; related to insulin, MAPK and mTOR pathway | *Ashby et al. (2016), Macedo et al. (2016), Shi et al. (2015)* | Down-regulated | Down-regulated but not significantly | Up-regulated but not significantly |
| Ame-mir-133 | Associated with the lipid loss in bees; participated in regulation of behavioral maturation in honey bees; up-regulated in 4-day-old queen larvae; related with MAPK pathway; down-regulated in *Ec* knock down bees | *Mello et al. (2014), Nunes et al. (2013), Shi et al. (2015)* | Down-regulated | No change | Down-regulated but not significantly |

Peer

**Table 4** (*continued*)

| miRNA id | Correlate | *Ref.* | Expression level in this study | | |
|---|---|---|---|---|---|
| | | | Q *vs.* V | C *vs.* Q | R *vs.* C |
| Ame-mir-14 | Caste determination; negatively related with *Ec* R expression and activity; up-regulated in 4-day-old queen larvae compared with 4-day-old worker larvae; related to insulin, MAPK, mTOR and Wnt pathway; down-regulated in *Ec* knock down bees; up-regulated in the activated ovaries | *Ashby et al. (2016), Macedo et al. (2016), Mello et al. (2014), Shi et al. (2015)* | Down-regulated | No change | Up-regulated but not significantly |
| Ame-mir-184 | Stable expression in active and inactive ovary; plays key roles in embryogenesis; the determination of the anteroposterior axis; embryo cellularization and stem cell determination; up-regulated in 4-day-old queen larvae compared with 4-day-old worker larvae; related to insulin pathway; down-regulated in *Ec* knock down bees; a miRNA in royal jelly and affect caste determination | *Guo et al. (2013), Macedo et al. (2016), Mello et al. (2014), Shi et al. (2015)* | Down-regulated | No change | Down-regulated but not significantly |
| Ame-mir-190 | Caste determination; immune-related | *Ashby et al. (2016)* | Down-regulated | No change | Down-regulated but not significantly |
| Ame-mir-252a | Up-regulated in 4-day-old queen larvae; up-regulated in the activated ovaries | *Macedo et al. (2016), Shi et al. (2015)* | Down-regulated | Down-regulated but not significantly | Up-regulated but not significantly |
| miR-263a | Associated with reproductive statuses; up-regulated in the inactive ovaries; down-regulated in *Ec* knock down bees | *Macedo et al. (2016), Mello et al. (2014)* | Up-regulated | Down-regulated but not significantly | Up-regulated but not significantly |
| Ame-mir-275 | *Vg* positive correlation; up-regulated in 4-day-old queen larvae; related to insulin and MAPK pathway | *Nunes et al. (2013), Shi et al. (2015)* | Down-regulated | Up-regulated | Up-regulated but not significantly |
| Ame-mir-276 | Associated with reproductive statuses; up-regulated in the inactive ovaries; down-regulated in *Ec* knock down bee; up-regulated in 4-day-old queen larvae compared with 4-day-old worker larvae | *Macedo et al. (2016), Mello et al. (2014), Shi et al. (2015)* | Down-regulated | Up-regulated | No change |
| Ame-mir-279 | Caste determination; immune-related; down-regulated in *Ec* knock down bees | *Ashby et al. (2016)* | Up-regulated | No change | Down-regulated but not significantly |
| Ame-mir-2796 | Participated in regulation of behavioral maturation in honey bees | *Nunes et al. (2013)* | Down-regulated | Down-regulated but not significantly | Down-regulated but not significantly |
| Ame-mir-279a-3p | Up-regulated in the activated ovaries | *Macedo et al. (2016)* | Down-regulated | Up-regulated but not significantly | Down-regulated but not significantly |

Chen et al. (2017), *PeerJ*, DOI 10.7717/peerj.3881

**Table 4** (*continued*)

| miRNA id | Correlate | *Ref.* | Expression level in this study | | |
|---|---|---|---|---|---|
| | | | Q *vs.* V | C *vs.* Q | R *vs.* C |
| Ame-mir-279b-3p | Up-regulated in the activated ovaries | *Macedo et al. (2016)* | Up-regulated | No change | Down-regulated but not significantly |
| Ame-mir-2944-3p | Up-regulated in the activated ovaries | *Macedo et al. (2016)* | Up-regulated | Down-regulated but not significantly | Up-regulated but not significantly |
| Ame-mir-305 | Down-regulated in *Ec* knock down bees | *Mello et al. (2014)* | Down-regulated but not significantly | Up-regulated | Up-regulated but not significantly |
| Ame-mir-306 | Associated with reproductive statuses; up-regulated in the activated ovaries; targets ATPsyn-beta-PA; down-regulated in *Ec* knock down bees | *Macedo et al. (2016)*, *Mello et al. (2014)* | Up-regulated | Down-regulated but not significantly | Down-regulated but not significantly |
| Ame-mir-315 | Caste determination; modulates tissue patterning and cell differentiation | *Ashby et al. (2016)* | Up-regulated | Down-regulated but not significantly | Down-regulated but not significantly |
| Ame-mir-316 | *Vg* negative correlation; related to Wnt pathway; down-regulated in *Ec* knock down bees | *Mello et al. (2014)*, *Nunes et al. (2013)* | Down-regulated | Up-regulated but not significantly | No change |
| Ame-mir-317 | Associated with reproductive statuses; up-regulated in the activated ovaries; related to insulin pathway; down-regulated in *Ec* knock down bees | *Macedo et al. (2016)*, *Mello et al. (2014)* | Down-regulated | Up-regulated but not significantly | No change |
| Ame-mir-31a | *Vg* negative correlation; associated with reproductive statuses; up-regulated in the inactive ovaries | *Macedo et al. (2016)*, *Nunes et al. (2013)* | Down-regulated | Up-regulated but not significantly | Down-regulated but not significantly |
| Ame-mir-33 | Caste determination; immune-related | *Ashby et al. (2016)* | Down-regulated | Up-regulated but not significantly | No change |
| Ame-mir-3718a | *Vg* negative correlation | *Nunes et al. (2013)* | Down-regulated | No change | Up-regulated but not significantly |
| Ame-mir-375 | Up-regulated in 4-day-old queen larvae; related to MAPK pathway | *Shi et al. (2015)* | Down-regulated | No change | Up-regulated but not significantly |
| Ame-mir-6001-3p | Up-regulated in 4-day-old queen larvae | *Shi et al. (2015)* | Down-regulated | Not detected | Up-regulated but not significantly |
| Ame-mir-71 | Participates in specific steps of the insulin/insulin-like signaling pathway | *Macedo et al. (2016)* | Down-regulated but not significantly | Up-regulated | Up-regulated but not significantly |

Chen et al. (2017), *PeerJ*, DOI 10.7717/peerj.3881

**Table 4** (*continued*)

| miRNA id | Correlate | Ref. | Expression level in this study | | |
|---|---|---|---|---|---|
| | | | Q *vs.* V | C *vs.* Q | R *vs.* C |
| Ame-mir-8 | Caste determination; immune-related; 20E and JH response miRNA; related to Wnt pathway; up-regulated in the activated ovaries | *Ashby et al. (2016)*, *Macedo et al. (2016)*, *Shi et al. (2015)* | Down-regulated | Up-regulated | No change |
| Ame-mir-92a | *Vg* negative correlation; participated in regulation of behavioral maturation in honey bees; associated with reproductive statuses; down-regulated in *Ec* knock down bees | *Macedo et al. (2016)*, *Mello et al. (2014)*, *Nunes et al. (2013)* | Up-regulated | Down-regulated but not significantly | No change |
| Ame-mir-92b | Up-regulated in the activated ovaries; related to insulin, MAPK and mTOR pathway; down-regulated in *Ec* knock down bees | *Macedo et al. (2016)*, *Mello et al. (2014)*, *Shi et al. (2015)* | Up-regulated | Down-regulated | Up-regulated but not significantly |
| Ame-mir-993 | Related to insulin pathway | *Shi et al. (2015)* | Down-regulated | Up-regulated but not significantly | Down-regulated but not significantly |
| Ame-mir-996 | Related to insulin pathway | *Shi et al. (2015)* | Down-regulated but not significantly | Up-regulated | Down-regulated but not significantly |

*al., 2014*; *Shi et al., 2015*). Furthermore, studies in *Dorsophlia* found that loss of miRNA-184 induced loss of egg production (*Iovino, Pane & Gaul, 2009*). In our trial, miRNA-184 was down-regulated in mated queens but not in virgin queens. Thus, it can be speculated that miRNA-184 could be a candidate marker for oviposition of the honey bee queen.

In addition, a positive correlation was observed between a set of three miRNAs (miR-263a, miR-2944-3p and miR-92b) and the oviposition status. MiR-263a and miR-92b were found to be involved in neuronal development and affected caste determination in honey bees (*Ashby et al., 2016*), which played important roles in reproductive activities (*Heifetz et al., 2014*). When the queen's oviposition is activated or regulated, the neuronal activity and excitability increases. Therefore, we deduced that miR-263a, miR-2944-3p and miR-92b might be associated with the neuronal development and they further affect oviposition activation and regulation.

Furthermore, as shown in Table 4, many miRNAs, which respond to *Ec*, *JH* and vitellogenin (*Vg*), showed significant changes in their expressions. *Ec*, *Vg* and *JH* are among the most important hormones in regulating reproductive activities in honey bees (*Lago et al., 2016*; *Nunes et al., 2013*; *Oxley & Oldroyd, 2010*). Particularly, *Vg* serves as a yolk precursor in egg development and affects oviposition in almost all oviparous species (*Downs, Mosey & Klinger, 2009*). Changes of expressions of miRNAs in our study may regulate or be regulated by *Ec*, *Vg* and *JH*, and further affect ovary activation and/or oviposition.

## Key roles in the lncRNA-miRNA-mRNA network

In the network constructed with miRNAs, mRNAs and lncRNAs, we found that some genes served as bridges linking different miRNAs, four of which (gene id: 408284, 408609, 409587, and 409152) acting as miRNA targets linked let-7, miR-100, miR-12, miR-14, miR-316 and miR-996. Two of them are worthy of noting here. One was coiled-coil domain-containing protein 93 (*CCDC93*, id: 408609). *Oh et al. (2011)* found that *CCDC93* regulated the expression level of cyclin B1 (*CycB1*), a cyclin gene in human cells. Our results showed that *CycB3*, another cyclin gene, was localized in the QTL region for ovary size, which indicated that *CCDC93* may interact with cyclin genes and further affect oviposition. The other was heat shock 70 kDa protein cognate (*Hsc70-3*, id: 409587). The interaction between *Hsc70* and *Hsp90* was reported previously (*King et al., 2001*). More importantly, the Hsc70/Hsp90 chaperone machinery is responsible for loading small RNA duplexes into Argonaute proteins, which are critical to small silencing RNAs—small interfering RNAs (siRNAs) or microRNAs (miRNAs)—direct posttranscriptional gene silencing of their mRNA targets (*Iwasaki et al., 2010*). Therefore, *Hsc70* is essential for miRNAs to implement their impact on the expression of target mRNAs. Our results confirm similar findings showing that *Hsc70-3* acted as a target of miRNA and served as a bridge linking different miRNAs in the lncRNA-miRNA-mRNA network, which highlight its role in the interaction among different RNAs in oviposition. Taken together, we can conclude that both *CCDC93* and *Hsc70-3* play important roles in the network and further affect gene expressions in oviposition.

## CONCLUSIONS

In the present study, lncRNAs, mRNAs and miRNAs expression profiles were evaluated and compared during ovary activation and dynamical oviposition process in honey bees. Bioinformatic analyses suggest that some lncRNAs, miRNA and genes are involved in important biological processes associated with oviposition activation and regulation. Additionally, lncRNA-miRNA-mRNA network revealed the potential interactions among different RNAs. Moreover, candidate genes or RNAs for oviposition were identified, which are particularly attractive for further in-depth studies.

## ACKNOWLEDGEMENTS

The authors thank Wei Feng, who is the beekeeper in the apiary, for his assistance in beekeeping.

### Funding

This study was supported by the China Postdoctoral Science Foundation funded project (2015M581222), the Agricultural Science and Technology Innovation Program (CAAS-ASTIP-2015-IAR) and the earmarked fund for Modern Agro-Industry Technology Research System (CARDS-45-KXJ1). The funders had no role in study design, data collection and analysis, decision to publish, or preparation of the manuscript.

### Grant Disclosures

The following grant information was disclosed by the authors:
China Postdoctoral Science Foundation: 2015M581222.
Agricultural Science and Technology Innovation Program: CAAS-ASTIP-2015-IAR.
Modern Agro-Industry Technology Research System: CARDS-45-KXJ1.

### Competing Interests

Ce Ma and Qian Lu are employees of Novogene Co., LTD.

### Author Contributions

- Xiao Chen conceived and designed the experiments, performed the experiments, analyzed the data, contributed reagents/materials/analysis tools, wrote the paper, prepared figures and/or tables, reviewed drafts of the paper.
- Ce Ma analyzed the data.
- Chao Chen and Qian Lu contributed reagents/materials/analysis tools.
- Wei Shi reviewed drafts of the paper.
- Zhiguang Liu, Huihua Wang and Haikun Guo take care of the honey bees used in this study.

### Data Availability

  GEO: GSE93028.

## Supplemental Information

Supplemental information for this article can be found online at http://dx.doi.org/10.7717/peerj.3881#supplemental-information.

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
