# Peer review of "Integration of lncRNA–miRNA–mRNA reveals novel insights into oviposition regulation in honey bees"

_PeerJ, doi:10.7717/peerj.3881_

## Round 0.1 · original submission · Major Revisions

Please heed all the comments from the two reviewers when considering revising the manuscript.

·

Basic reporting

The manuscript is flawed because it did not make clear the major question and the hypothesis of the study, neither in the abstract nor in the introduction and/or discussion sections.

The authors did not cite literature already published in Apis mellifera, which make important relationships for the robustness and support of this manuscript:

(1) ovarian activity and microRNAs (Macedo et al., 2016 - Insect Mol Biol. 25(3):216-26.);

(2) vitellogenin and microRNAs (Nunes et al., 2013 - J Exp Biol. 216(Pt 19):3724-32.);

(3) EcR-mRNA-microRNAs (Mello et al., 2014 - Front Genet. 5:445.);

(4) caste and microRNAs (Guo et al., 2013 - PLoS One. 8(12):e81661.; Shi et al., 2014 - Apidologie. 46:35);

(5) ovary, caste, mRNA, long noncoding RNAs (Humann and Hartfelder, 2011, Insect Biochem Mol Biol. 41(8):602-12.; Humann et al., 2013. PLoS One. 8(10):e78915.; Jayakodi et al., 2015 - BMC Genomics. 16:680.; Lago et al., 2016 - Insect Biochem Mol Biol. 79:1-12.).

Figures, tables and legends are not clear and should be improved. For example: What does A, B and C mean in Figure 1?

Experimental design

In my opinion, the research would fill an identified knowledge gap if question was well defined (but it is not well defined) and if data were appropriately analyzed. Methods are not described with sufficient information to be reproducible by another investigator. For example:

(1) how lncRNA's target genes were predicted?

(2) how many genes were validated by qPCR (5 lncRNA as in line 165 and 298? or 9 as in "result sheet" of Table S1? or 10 (used actin as reference gene?) as in "primer sheet" of Table S1?

(3) What does microarray mean in line 170?

(4) Line 152: What does "trans role is to identify each other by the expression level" mean?

Validity of the findings

Sequencing data appear to be of sufficient quality and preliminary analyzes were well performed. However, the experimental design of comparisons to know whether a gene is differentially expressed is unclear. What experimental design was used: reference design? loop design? saturated design? others? They used DESeq to compare any two groups (lines 132-135) / discriminate between two comparison groups (lines 179-180), however I'm not convinced that this is the best form of analysis. I would have agreed with authors if they had been performed all possible pairwise comparisons. If they have done all them, this is not clear from the text of the manuscript.

Finally, the authors performed an unsupervised hierarchical clustering analysis, which is fine (great job!). But they need to clarify in more detail about the use of each approach and the parameters used in each of them. How lncRNA's targets were predicted? About the construction of networks, the authors assumed two expression patterns: (1) up-regulated lncRNA / mRNA versus down-regulated miRNA or (2) down-regulated lncRNA / mRNA versus up-regulated miRNA. In what literature have the authors based themselves on assuming such premises or supporting such expression patterns?

In my view, conclusion are not well stated and not limited to supporting results. The queen's reproduction regulation was not elucidated (lines 375-378). The conclusion does not present the answer of a scientific question (knowledge gap). There are many speculations, but they have not been identified as such.

Additional comments

Dr. Xiao Chen and colleagues investigated the expression of genes (mRNA, miRNA and lncRNA) in queen ovaries undergoing four different contexts () in order to understand the molecular events associated with prolificacy. Despite the quality data, the written information in the manuscript is vague, generic, confusing or inconclusive. In general, it seems more a making-list + long descriptions than the expected interpretive deepening of the data.

Below, some points and suggestions:

1 - Prolificacy is an argument used in the abstract, in the beginning of the introduction and in line 340 of the discussion. I suggest that the authors use the data generated to explain this term "molecularly".

Throughout the text, it is very confusing to understand in which situation a gene is up- or down-regulated. The authors should make clear which is the reference sample, and always declare something like that: "up-regulated in situation A compared to situation B (or situations B, C and D)", for example.

It is not clear from the text what the new insights are. The found GO terms are too vague and / or generic (for example, system and organism development / metabolic process / etc.) so that the knowledge can not be advanced/deepen.

Line 129: Did miRNA target prediction use only 3'UTRs of target genes? Please, clarify it.

Lines 160-170: About Quantitative qPCR, how many genes were really validated? I observed primers designed for Gene ID 726542 (histone H3) but no results. Also, if authors used oligodT primer, it means that all validated genes are polyadenylated, are not they?

Line 290: network of ovaries activation or network of oviposition recovery? Please, clarify it.

Lines 331-352: microRNAs such as let-7, miR-316, miR-252, miR-1, miR-133, miR-375, miR-31a, miR-92a, miR-3049, miR-3718 should be better (Nielsen et al., 1994), and the results obtained in the present study were similar to those reported in the present study (Macedo, et al., 1991, Insec Mol Mol Biol 25 (3): 216-26). Exp Biol. 216 (Pt. 19): 3724-32.).

In addition, please also observe all the criticisms / comments / suggestions presented in the sections above: 1. Basic reporting, 2. Experimental design, 3. Validity of the findings.

Reviewer 2 ·

Basic reporting

The English in the manuscript needs to be improved. In many instances the language does not make the meaning of the sentences unclear, but it would still be improved considerably by having a native English speaker review the manuscript before resubmission.

An example is lines 346-347: “Ame-miR-315 down regulated in ovaries activation and target 253 mRNAs in the network.” This should say: “Ame-miR-315 is down regulated in the ovary activation samples and targets 253 mRNAs in the network” It could also say “… regulated in activated ovaries and targets…”
Along these lines, while “ovaries” is the plural of ovary, when referring to a sample type (e.g. ovary activation) it should be singular because it is a category. This is done in the introduction and the figure legends, but not in the rest of the manuscript.

There are some cases in which it is unclear what the meaning of a sentence is.
Line 152: “Trans role is lncRNA to identify each other by the expression level.” This sentence doesn’t make sense. A google search reveals that this is used multiple times, virtually verbatim in several publications over the last 3 years. These other papers seem to be making the distinction between cis and trans effects that lncRNAs can have (acting on neighboring genes, cis, or acting on distant genes, trans). However, “identify each other by the expression level” is completely unclear. How does a lncRNA “identify” something based on expression level? The remainder of the paragraph says that the authors looked for correlations between the expression levels of lncRNAs and coding genes. If the first sentence is meant to state this, it would be better stated as something like: “The potential trans role of lncRNAs (acting on non-neighboring genes) can be assessed by correlating expression levels between lncRNAs and mRNAs.”
Importantly, this sentence and the sentence following it are nearly identical to those used by other authors, particularly Lu et al., (2016). I understand that some technical phrases are difficult to rewrite and I am not accusing the authors or plagiarism, but it is important to rework these ideas into your own words. This case is particularly poignant because the sentence is completely ambiguous and has been passed down over several papers.

I have also found the interpretation to be difficult due to potentially ambiguous use of the term “target.” Targeting has a directionality to it and finding that X targets Y doesn’t mean that Y targets X.
Lines 74-75: “…lncRNAs can also be directly targeted by miRNAs for cleavage.”
This line is clear and is directly supported by the citations, they show that miRNAs target the lncRNAs (miRNA acts upon the lncRNA), not the other way around.
Lines 155-156: “…lncRNAs and mRNAs which linked by the same target miRNAs, with…” In this case, the sentence states that the lncRNAs and mRNAs both target (act upon) one particular mRNA. As far as I am aware this is not the direction in which this works, mRNAs and lncRNAs are targeted by miRNAs (miRNAs act upon the others) and not the other way around. The Fan et al., (2015) paper supports this, the lncRNAs can influence mRNA expression level by interfering with the miRNAs that normally act upon that mRNA, but they interfere by being targeted by the miRNAs, not by targeting the miRNAs.
I think that the Fan et al., 2015 reference does a good job in making this clear in their language (“lncRNAs acting as miRNA targets”).

The article contains many other grammatical, spelling, and formatting problems. Some of these should be caught by spell-checking software (e.g. line 156 “up-regualted”). There are a number of cases in which spaces are omitted (lines 126, 128, Figure 1 and Table 2 legends). These may be due to pdf formatting, but should be checked. Similarly, in the note of the Table 2 legend, the “Q” group seems to have a copy/paste error “Q, group of virgin queens normal oviposition queens;”.

Results section, Lines 234-296. These 6 paragraphs are tedious to read, if only because they are largely the same format and just report the two main patterns (up/down/up or down/up/down) in the three ovary states (activated, inhibited, recovery). Could these be incorporated into a table rather than being written out? If this can be done, it would make the results, and therefore the paper, much more concise and would make it easier to focus.
Line 259-260: What is meant by “lncRNAs target miRNAs from 2 to 3” or “miRNAs target mRNAs from 9 to 18”? Does it mean that the lncRNAs have from 2 to 3 target miRNAs? (however, see note above about use of “target”) If this is the case, then it should be spelled out. E.g. lncRNAs act as targets for 2 to 3 miRNAs.

Figure 1: Why does the mRNA heat map show all 3 samples for each treatment, but the others only show a single column for each treatment?

Supplemental files: I appreciate the supplemental files as it makes it much easier to assess the results. However, it would be helpful to have a single file that includes all the details about the supplemental tables, including explanations of headers in those files. The header explanations could be included as a separate sheet in the table workbooks.

I didn’t look at every entry in every supplemental file, but I found two errors. The first sheets in Supplemental Tables 3 and 5 have rows that only contain FBgn names and they look like a pasting error (S3 row 2687, S5 rows 1644 and 1646). The rest of the files should be checked for these types of errors.

Experimental design

This work is original primary research that fits within the aims and scope of PeerJ. Overall, the experimental design and data collection appear rigorous. However, the methods need to be more clearly spelled out. I lay out some questions that should be addressed in a new version.

Lines 151-159. What are some of the specifics of constructing a network?
1) Line 155-156: This states that lncRNAs and mRNAs were linked by the same target miRNA (presumably they are targeted by the same miRNA). However, lines 224-229 states that the network was constructed based on the correlation analysis, which looked just at expression levels. Which is the case?
2) What does it mean to have correlated expression? Do they have to have the same pattern (e.g. both upregulated), or do they have to have the same magnitude within a pattern (e.g. each shows a 3.5 fold upregulation)?
3) How were targets determined? Line 154 says that miRNA targets were determined using miRanda, but what parameters were used?
4) Why wasn’t GSTAr.pl run like Fan et al., 2015? That seems like an important analysis to determine whether a lncRNA is a decoy vs a target of a miRNA. While the manuscript isn’t explicitly about decoy vs target, the authors ultimately want to conclude that lncRNAs are acting to regulate mRNAs through this network, and they seem to rely on the lncRNAs acting as decoys.
5) Lines 93-96: Were these ovaries from a single queen? It seems like this is the case, but explicitly stating it would be helpful.
6) Lines 131-138: It is never explicitly stated what comparison is used for the DE analysis. Based on the categories of DE used in Figs 2-4, it seems that over- and under-expression are based on comparisons with the virgin queens (i.e. overexpression means: Q>V, C>V, R>V and underexpression means: Q<V, C<V, R<V). This should be made clear in the methods.

Validity of the findings

Apart from the language issues and making the results more concise, my chief concern is the network analysis interpretation. Here is my understanding based on the manuscript:

RNAs were considered to be in the same network if they showed one of the two patterns lncRNA-miRNA-mRNA (up-down-up or down-up-down). lncRNA and mRNA seem to also be linked by being targeted by the same miRNA (line155-156, see comment in experimental design section).

I think that this network analysis makes sense when considering the expression levels, but I think it goes awry when trying to look at functional classifications because it begs the question (i.e. it is circular). GO terms and KEGG pathways were determined for lncRNAs and miRNAs by looking at the functions of their targets, which are mRNAs. These mRNAs should be already in the network, since the network is defined in part by the shared targets (one miRNA targeting a lncRNA and an mRNA). If this is correct, then that means that any functional analysis of lncRNAs or miRNAs should find considerable overlap between these RNAs and the mRNAs. This leads to spurious conclusions, like lines 204-205: “The result revealed that the lncRNAs and mRNAs interacted with each other and promoted the ovaries activation and the subsequent oviposition process.” This is exactly what you would expect based on the analysis regardless of any actual interaction. I’m not saying they didn’t interact, but the only evidence of that is the shared miRNAs and the expression pattern, the functional analysis adds no information.
I think that the functional analysis can stay, if done solely at the mRNA target level and not attempting to compare it with the lncRNA or miRNA. This will still be helpful for understanding what the genes in these networks do, but it can’t be used to support the existence of the network itself.

More broadly speaking, I think the manuscript can be improved with a bit more discussion of what these processes mean for ovary activation and development. There are a lot of GO terms and KEGG pathways stated in the results and it would be helpful to have a more in depth biological discussion. The current manuscript relies heavily on the network analysis, but I think this should play a smaller supporting role. It seems that the main story should be about the biological processes involved in ovary regulation and the network analysis is a method of finding what functions these ncRNAs might have.

Additional comments

In the manuscript, “Integration of lncRNA–miRNA–mRNA reveals novel insights into reproductive regulation in honey bees,” Chen et al. report on expression levels of long-noncoding RNAs, miRNAs, and mRNAs in the ovaries of queen honey bees in four conditions of egg laying. These are: virgin queens not laying eggs, normal queens laying eggs, caged queens that are restricted from laying eggs, and recovering queens that were restricted but have been allowed to begin laying eggs again. They analyzed expression regulatory networks by comparing expression levels and potential targets between these categories of RNAs and by comparing the predicted functions of these genes (using GO and KEGG analyses). I think that this project is very exciting, that the experiments and data are solid, and that the results will be very useful for our understanding of honey bee reproductive biology and how it is regulated. However, I have concerns about the analyses and presentation of the work. I recommend that the manuscript undergo major revisions to address these concerns and to make the manuscript more concise.

---

## Round 0.2 · Major Revisions

Besides the other comments, please pay special attention, when submitting a revised version of the manuscript, on clarification and improvement of the language, since, as it is now, the English hinders correct understanding of the paper, and this has been noted multiple times by both reviewers.

·

Basic reporting

The manuscript was substantially improved. However, language still needs many tweaks and improvements. Several sentences remain vague, confusing or meaningless. This makes it difficult to read and consequently to understand the manuscript.

Experimental design

Language needs to be improved to allow full understanding of the text.

Validity of the findings

Language needs to be improved to allow full understanding of the text.

Additional comments

Language needs to be improved to allow full understanding of the text.

Also, please provide nucleotide sequences (or GenBank accession number) of all data named as TCONS_?

How can authors know that many TCONS_ sequences are genuine lncRNA?

To my knowledge, 20E is a symbol used for 20-hydroxyecdysone.

Reviewer 2 ·

Basic reporting

1 The English has been improved in the manuscript, removing many ambiguous sentences. However, there are still many grammatical errors throughout.

2 Lines 42-44: The first sentence makes a very broad statement with no citation included. What are the biological constrain(t)s that are going to be ameliorated? If it’s the complex genetic architecture, then this is circular reasoning. These sentences together state that the way to ameliorate the problem of complex genetic architecture of reproduction is to better understand the genetics of reproduction.

3 Lines 65-66: This is another very broad statement with no citation.

4 Figures and tables: The network figures (Figures 2 and 3) are large, crowded and not readable. If these kinds of network figures are included, they should be scaled down or altered to guide the reader through the network. As it stands, they are simply too complex to serve as a main figure. If the full network needs to be included, then it should go in the supplement. Figure 4 is still very large, but this one at least shows that many of the genes in the pathways are DE in this study.
Table 3 isn’t a table, it’s a list. Again, if the authors feel this is necessary then it should be in the supplement. In that case, it can be converted into a useful format (spreadsheet) so that each gene has it’s own row, so that readers can easily sort or copy IDs.

5 In my previous review, I brought up concerns about performing functional analyses (GO and KEGG) on the lncRNA and miRNA, because those analyses rely on using networked partners of those RNAs (they use the mRNAs). This was point #19 in the previous review. While the authors did clarify their methods for making the network and don’t use the functional analyses to support the existence of a network, they still performed the functional analyses on both lncRNAs and miRNAs and proceed to discuss how similar the results of those analyses are to the results of the mRNA analysis. I don’t see any way that the results could not overlap because the three RNA types have already been shown to be in a network (based on expression correlation). The function of the lncRNAs and miRNAs are defined by the mRNAs with which they interact, so of course they overlap. If there is something I’m missing on why this analysis was performed and why it’s discussed, then that needs to be made very clear in the manuscript. It currently is not.

6 Lines 329, 365, and 431: None of these sentences should use the term “proved”, these are scientific hypotheses that have been supported.

Experimental design

The authors have improved the description of the methods significantly.
Knowledge Gap. I think that the work attempts to fill a gap in our knowledge of the genetic regulation of oviposition in honey bees. I’m not so sure that they spell out that knowledge gap very well (see comments above regarding lines 42-44).
The manuscript fits within the aims and scope of the journal.

Validity of the findings

I appreciate the addition of the QTL for ovariole number, I think it improves the strength of the current conclusions. However, the language makes it hard to determine what are new conclusions and what are conclusions from previous work. Lines 179-181 make it sound like this was a QTL analysis performed for this manuscript, but it was the work of Linksvayer and Graham. I don’t think this is intentional, but simply stating that it was “previously localized to…” would clarify.
This same issue comes up a lot in the discussion. I appreciate that the authors are discussing their findings in relation to what is known, but it is not always clear when new conclusions are being made and when they are simply stating the previous conclusions. The use of the past tense also makes this more difficult (e.g. lines 587-588, “Nej was considered…”)

Additional comments

I appreciate the fuller discussion of the the genes in relation to previous work, however I still feel that the manuscript could be improved by having a common thread for the reader to focus on throughout the discussion. In some ways the discussion feels like a list of facts about these pathways without being tied back together into a proper conclusion. I understand that a lot of this work results in new candidate genes to be further investigated, but more comprehensive conclusions would help to guide the reader.

---

## Round 0.3 · Minor Revisions

Please revise the English as it is still not at an adequate level for readership.

·

Basic reporting

no comment

Experimental design

no comment

Validity of the findings

no comment

Additional comments

The authors adequately addressed all of my questions/suggestions (as well as those of the other reviewer), clarifying them satisfactorily and significantly improving the quality (and also the language) of the manuscript. The study increases our understanding of the molecular relationships between the non-protein-coding layers of bee genome and the reproductive biology of Hymenoptera. I recommend that the revised version of the manuscript be accepted for publication at PeerJ.

---

## Round 0.4 · accepted · Accept

While in production, please check one last time the English, as there are still some few mistakes.